# On the Robustness of deep learning-based MRI Reconstruction to image transformations

**Jinghan Jia**
Michigan State University
`jiajingh@msu.edu`

**Mingyi Hong**
University of Minnesota
`mhong@umn.edu`

**Yimeng Zhang**
Michigan State University
`zhan1853@msu.edu`

**Mehmet Akçakaya**
University of Minnesota
`akcakaya@umn.edu`

**Sijia Liu**
Michigan State University
`liusiji5@msu.edu`

## Abstract

Although deep learning (DL) has received much attention in accelerated magnetic resonance imaging (MRI), recent studies show that tiny input perturbations may lead to instabilities of DL-based MRI reconstruction models. However, the approaches of robustifying these models are underdeveloped. Compared to image classification, it could be much more challenging to achieve a robust MRI image reconstruction network considering its regression-based learning objective, limited amount of training data, and lack of efficient robustness metrics. To circumvent the above limitations, our work revisits the problem of DL-based image reconstruction through the lens of robust machine learning. We find a new instability source of MRI image reconstruction, *i.e.*, the lack of reconstruction robustness against spatial transformations of an input, *e.g.*, rotation and cutout. Inspired by this new robustness metric, we develop a robustness-aware image reconstruction method that can defend against both pixel-wise adversarial perturbations as well as spatial transformations. Extensive experiments are also conducted to demonstrate the effectiveness of our proposed approaches.

## 1 Introduction

Medical image reconstruction is a known problem in magnetic resonance imaging (MRI). The key challenge lies in solving an inverse problem so as to enable the reconstruction of a high-quality medical image from sub-sampled measurements that are readily be obtained from medical devices [Lustig et al., 2008]. The conventional approach for solving the MRI image reconstruction problem commonly resorted to sparse signal processing techniques [Lustig et al., 2008, Yang et al., 2010, Huang et al., 2011], *e.g.*, compressed sensing. The key enabling technique is to leverage sparsity-inducing optimization [Bach et al., 2011], *e.g.*, the least-squared problem penalized by $\ell_1$ norm. In contrast to the direct optimization-based approaches, an increasing amount of recent works proposed the deep learning (DL)-based MRI image reconstruction methods [Zhu et al., 2018, Liang et al., 2020, Sriram et al., 2020]. With aid of DL models, the problem of image reconstruction is then solved as supervised learning problem in which the reconstruction error between the 'predicted' high-quality image and the train-time 'reference' image is minimized. In this work, our research falls into the category of DL-based MRI image reconstruction.

The DL-based image reconstruction approaches can be roughly divided into three types. First, U-Net [Ronneberger et al., 2015] and its many variants [Han and Ye, 2018, Lee et al., 2018] follow the Encoder-Decoder based network architecture. Second, learning-to-optimize (L2O), which leverages a recurrent neural network (RNN) to mimic an iterative optimization algorithm in problem solving

2022 Trustworthy and Socially Responsible Machine Learning (TSRML 2022) co-located with NeurIPS 2022.

[Chen et al., 2017], was adopted to unroll the optimization flow of sparse signal recovery for image reconstruction [Yang et al., 2016, Aggarwal et al., 2018, Hosseini et al., 2020] Third, in addition to supervised learning, state-of-the-art unsupervised learning approaches were leveraged to tackle medical image reconstruction in the scarce data regime [Hu et al., 2021, Fabian et al., 2021].

In spite of the rapid growth in DL for medical image reconstruction, nearly all existing works focused on tackling the problem of enhancing image *reconstruction accuracy*. However, a series of recent works [Antun et al., 2020, Zhang et al., 2021] showed that the DL-based image reconstruction network suffers a *weakness*: the lack of adversarial robustness. Specifically, it has been shown in Antun et al. [2020] that an imagery input, at the presence of adversarial perturbations (which are designed via adversarial learning, known as adversarial attack generation), can significantly hamper the image reconstruction accuracy even if these perturbations are unnoticeable to human eyes. We refer readers to Fig. 2(b) for a reconstructed image example using an DL-based approach when facing adversarial input perturbations. Prior to the finding of the lack of adversarial robustness in *image reconstruction*, the problem of adversarial learning has been extensively studied in the domain of *image classification* [Goodfellow et al., 2014, Carlini and Wagner, 2017, Croce and Hein, 2020]. To defend such adversarial perturbations, the min-max optimization-based adversarial training (AT) [Madry et al., 2017] provides a principled defensive method to robustify image classifiers. By contrast, few work studied the problem of improving adversarial robustness for deep image reconstruction networks, except Raj et al. [2020], Calivá et al. [2020] which were mainly built upon AT. Different from the existing work, our **contributions** are summarized below.

• We show that in addition to adversarial perturbation, deep MRI image reconstruction models are also vulnerable to tiny input spatial transformations.

• We develop a generalized adversarial training (GAT) approach that promotes robustness against both adversarial input perturbations and input spatial transformations.

• We demonstrate the effectiveness of GAT on the MRI dataset [Aggarwal et al., 2018], showing our improvement over the conventional standard training and adversarial training.

## 2 Preliminaries

In this section, we provide a brief background on the problem of accelerated MRI acquisition, and the deep learning (DL) based MRI reconstruction method.

**Accelerated multi-coil MRI acquisition**   In MRI , measurement of a patient's anatomy is acquired in the Fourier-domain, also known as the $k$-space, through receiver coils. In the setup of multi-coil MRI acquisition, each coil will capture a different region of a targeted image, and will be assigned with a complex-valued sensitive map, noted by $\mathbf{S}_i$. As a result, the $k$-space measurement $\mathbf{y} \in \mathbb{C}^n$ of a complex-valued ground truth image $\mathbf{x}^* \in \mathbb{C}^n$ is given by :

$$\mathbf{y}_i = \mathbf{P}_\Omega \mathcal{F} \mathbf{S}_i \mathbf{x}^* + \mathbf{n}_i, \quad i = 1, ..., N, \tag{1}$$

where $\mathbf{N}$ is the number of coils, $\mathbf{P}_\Omega$ is a binary sampling mask that will be introduced later, $\mathcal{F}$ is the two-dimensional Fourier-transform, and $\mathbf{n}_i \in \mathbb{C}^n$ is the additive noise arising in the measurement process. In (1), the use of $\mathbf{P}_\Omega$ is for the purpose of *accelerating* the multi-coil MRI acquisition, since obtaining the full-sampled measurements in the $k$-space is quite time-consuming. The 0s in $\mathbf{P}_\Omega$ represents the frequencies that will not be sampled. We refer readers to Deshmane et al. [2012] for more details on multi-coil MRI acquisition. Eventually, the multi-coil MRI acquisition process can be formulated as the forward mapping below [Pruessmann et al., 2001]

$$\mathbf{y}_\Omega = \mathbf{A}_\Omega \mathbf{x}^* + \mathbf{n} \tag{2}$$

where $\mathbf{y}_\Omega$ denotes the sub-sampled multi-coil $k$-space measurements, $\mathcal{A}_\Omega$ denotes the forward encoding operator which concatenates $\mathbf{P}_\Omega \mathcal{F} \mathbf{S}_i$ across all coils.

**MRI reconstruction problem**   The goal of MRI reconstruction is to estimate the original image $\mathbf{x}^*$ from the $k$-space measurement $\mathbf{y}_\Omega$. The conventional solution to dealing with the MRI reconstruction problem is to treat the problem as an inverse problem and then calls the regularized least-squared method [Pruessmann et al., 2001, Sutton et al., 2003] . However, the state-of-the-art reconstruction performance is achieved by a learning-based approach, especially for using deep models. In this paradigm, we aim to learn a reconstruction network $f_{\boldsymbol{\theta}}$, parameterized by $\boldsymbol{\theta}$, under a training data set $\mathcal{D}$. In the training set, the data 'feature' is given by the sub-sampled image $\mathbf{z}$ acquired from the

$k$-space measurement directly, namely, $\mathbf{z} = \mathbf{A}_\Omega^H \mathbf{y}_\Omega$, and the data 'label' is given by the original image $\mathbf{x}^*$. The rationale behind using $\mathbf{z}$ rather than $\mathbf{y}_\Omega$ is that the mapping from $\mathbf{z}$ and $\mathbf{x}^*$ can be readily achieved using a denoising-based neural network [Aggarwal et al., 2018] as the input resolution and the output resolution remain the same. The deep learning (DL) based MRI reconstruction then seeks the optimal network parameters $\boldsymbol{\theta}$ by minimizing a certain training loss $\ell$ over the training dataset $\mathcal{D}$:

$$\min_{\boldsymbol{\theta}} \ \mathbb{E}_{(\mathbf{z}, \mathbf{x}^*) \in \mathcal{D}} \left[ \ell(f_{\boldsymbol{\theta}}(\mathbf{z}), \mathbf{x}^*) \right], \tag{3}$$

where recall that $f_{\boldsymbol{\theta}}(\mathbf{z})$ and $\mathbf{x}^*$ correspond to the network output image and the reference fully-sampled image, respectively. In practice, a normalized $\ell_1$-$\ell_2$ loss [Knoll et al., 2020a] is commonly used to specify $\ell$, which is given by

$$\ell(f_{\boldsymbol{\theta}}(\mathbf{z}), \mathbf{x}^*) = \frac{\|f_{\boldsymbol{\theta}}(\mathbf{z}) - \mathbf{x}^*\|_2}{\|\mathbf{x}^*\|_2} + \frac{\|f_{\boldsymbol{\theta}}(\mathbf{z}) - \mathbf{x}^*\|_1}{\|\mathbf{x}^*\|_1}. \tag{4}$$

Fig. 1 shows the input sub-sampled image $\mathbf{z}$, the reconstructed image obtained from the normally trained model $\boldsymbol{\theta}_{\mathrm{NT}}$ by solving problem (3), and the full-sampled reference image $\mathbf{x}^*$.



(a) input $\mathbf{z}$  (b) output $f_{\boldsymbol{\theta}_{\mathrm{NT}}}(\mathbf{z})$  (c) ground truth $x^*$

Figure 1: Example of multi-coil MRI image reconstruction using Model-based deep learning architecture [Aggarwal et al., 2018] under fastMRI knee [Knoll et al., 2020b] dataset by solving (3). We refer readers to Sec. 5 for more implementation details.

## 3 The Lack of Robustness for DL-Based MRI Reconstruction

In this section, we will demonstrate two vulnerabilities of the DL-based MRI reconstruction network to input 'perturbations'. First, we echo the finding of Antun et al. [2020] that DL-based MRI reconstruction is oversensitive to *adversarial input perturbations*, which are unnoticeable but carefully crafted to worsen the reconstruction accuracy. Second, beyond Antun et al. [2020], we find that DL-based MRI reconstruction also lacks robustness against *input spatial transformations*, *e.g.*, flipping, cutting-out, and rotation, even to a small degree. To the best of our knowledge, the vulnerability of DL-based MRI reconstruction to spatial transformations of inputs is found for the first time.

**Norm-constrained adversarial input perturbations**   Given a well-trained reconstruction network $f_{\boldsymbol{\theta}_{\mathrm{NT}}}$ from (3), the norm-constrained adversarial attack was first established by Antun et al. [2020] in the context of MRI image reconstruction. Following the same spirit of adversarial attack in image classification [Goodfellow et al., 2014], the adversarially perturbed input is formulated as $\mathbf{z} + \boldsymbol{\delta}$, where $\boldsymbol{\delta}$ denotes adversarial perturbations. The adversary then optimizes $\boldsymbol{\delta}$ to degrade the reconstruction performance. To this end, one can solve the optimization problem [Antun et al., 2020],

$$\max_{\|\boldsymbol{\delta}\|_\infty \leq \epsilon} \ell(f_{\boldsymbol{\theta}_{\mathrm{NT}}}(\mathbf{z} + \boldsymbol{\delta}), f_{\boldsymbol{\theta}_{\mathrm{NT}}}(\mathbf{z})), \tag{5}$$

where $\|\cdot\|_\infty$ denotes the $\ell_\infty$ norm, $\epsilon > 0$ is the perturbation budget, and $\ell$ signifies the reconstruction loss, *e.g.*, given by (4). The rational behind (5) is to learn $\boldsymbol{\delta}$ so as to enlarge the discrepancy between the model outputs at the perturbed input and the original input, respectively. To solve the optimization problem (5), the standard projected gradient descent (PGD) [Madry et al., 2017] is commonly used. In Fig. 2, we compare the image reconstruction results (when facing the benign input and the adversarial perturbed input, respectively) with the full-sampled reference image $\mathbf{x}^*$. To generate the adversarial perturbations $\boldsymbol{\delta}$, we use PGD to solve (5) with $\epsilon = 0.03/255$.

**Spatial transformation-based input perturbations.**   We next introduce a non-norm-constrained adversarial perturbations by leveraging input spatial transformations, *e.g.*, flipping, cutting-out, and rotation. We ask: *Will DL-based MRI reconstruction network be robust against heuristics-based input*

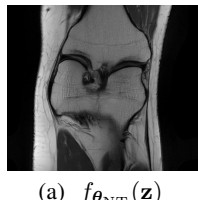 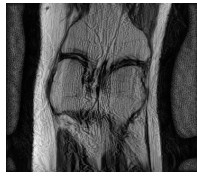 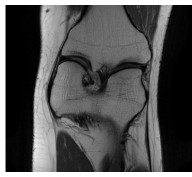

(a) $f_{\boldsymbol{\theta}_{\mathrm{NT}}}(\mathbf{z})$      (b) $f_{\boldsymbol{\theta}_{\mathrm{NT}}}(\mathbf{z} + \boldsymbol{\delta})$      (c) ground truth $\mathbf{x}^*$

Figure 2: Example of MRI image reconstruction based on the benign input $\mathbf{z}$ and the adversarial input $\mathbf{z} + \boldsymbol{\delta}$, versus the reference image $\mathbf{x}^*$.

*transformations?* Different from adversarial perturbations that are determined by solving (5), the input transformations are randomly generated. Let $T : \mathbb{C}^n \to \mathbb{C}^n$ denote a transformation operation, the perturbed input is then given by

$$\mathbf{z}' = T(\mathbf{z}). \tag{6}$$

Associated with (6), the desired reference image will become $T(\mathbf{x}^*)$. Given the normally trained network $f_{\boldsymbol{\theta}_{\mathrm{NT}}}$, we find that there exists a large discrepancy between $f_{\boldsymbol{\theta}_{\mathrm{NT}}}(T(\mathbf{z}))$ and its reference image $T(\mathbf{x}^*)$. An example is shown in Fig. 3, where a rotation of $180°$ is applied to the input image $\mathbf{z}$. Moreover, by comparing $f_{\boldsymbol{\theta}_{\mathrm{NT}}}(T(\mathbf{z}))$ with $f_{\boldsymbol{\theta}_{\mathrm{NT}}}(\mathbf{z} + \boldsymbol{\delta})$ in Fig. 2, we observe that DL-based MRI reconstruction lacks robustness against not only optimization-based adversarial perturbations but also heuristics-based input transformations.

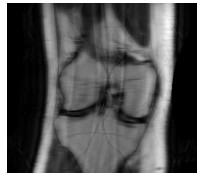 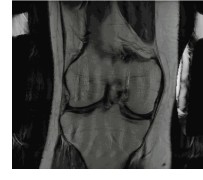 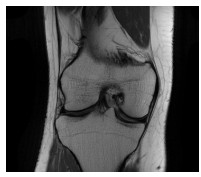

(a) input $\mathbf{z}'$      (b) $f_{\boldsymbol{\theta}_{\mathrm{NT}}}(T(\mathbf{z}))$      (c) ground truth $T(\mathbf{x}^*)$

Figure 3: Example of MRI reconstruction at the spatially transformed input (by rotating $180°$), the output image, and the reference image.

## 4 Towards Unified Robustness of MRI Reconstruction

In this section, we will develop novel training protocols to acquire robust DL-based MRI reconstruction networks against both adversarial perturbations and input transformations. To defend adversarial perturbations, the min-max optimization (MMO)-based adversarial training (AT) recipe will be used following the same spirit of robust image classification [Madry et al., 2017]. However, we will show that AT lacks efficacy to improve robustness against input transformations. Spurred by that, we will then propose a generalized AT approach that can incorporate data augmentations for 'free', namely, without needing additional data annotations.

**Adversarial training (AT)** AT adopts MMO as the algorithmic backbone, where the *worst-case (maximum)* training loss is *minimized* by incorporating a synthesized attack during model training. For robust image classification, AT has been poised the only effective defense method [Athalye et al., 2018] against the adversarial input perturbations. Based on MMO, problem (3) can be modified as

$$\min_{\boldsymbol{\theta}} \ \mathbb{E}_{(\mathbf{z}, \mathbf{x}_*) \in \mathcal{D}} \max_{\|\boldsymbol{\delta}\|_{\infty} \leq \epsilon} \left[ \ell(f_{\boldsymbol{\theta}}(\mathbf{z} + \boldsymbol{\delta}), \mathbf{x}^*) \right], \tag{7}$$

where the training loss is given by (4). Compared to (3), the attack generation problem (5) is infused into (7) as an inner maximization problem. To solve problem (7), the alternating gradient ascent-descent method [Madry et al., 2017] is commonly used. For ease of presentation, let $\boldsymbol{\theta}_{\mathrm{AT}}$ denote the model parameters obtained from (7). This is in contrast to $\boldsymbol{\theta}_{\mathrm{NT}}$ obtained by the normal training over (3). In Fig. 4, we demonstrate an example to compare the reconstruction performance of $f_{\boldsymbol{\theta}_{\mathrm{AT}}}(\mathbf{z} + \boldsymbol{\delta})$ and that of $f_{\boldsymbol{\theta}_{\mathrm{NT}}}(\mathbf{z} + \boldsymbol{\delta})$ when facing adversarial perturbations. Meanwhile, we show the performance of these different models against input transformations. As we can see, AT largely enhances the robustness of MRI reconstruction against adversarial input perturbations. However, it is insufficient to robustify the reconstruction network against the input transformations; see Fig. 4-(c).

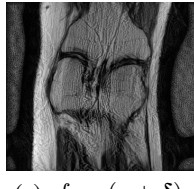
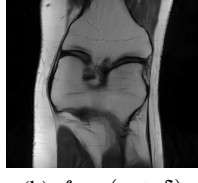
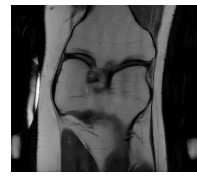

(a) $f_{\boldsymbol{\theta}_{\mathrm{NT}}}(\mathbf{z} + \boldsymbol{\delta})$      (b) $f_{\boldsymbol{\theta}_{\mathrm{AT}}}(\mathbf{z} + \boldsymbol{\delta})$      (c) $f_{\boldsymbol{\theta}_{\mathrm{AT}}}(T(\mathbf{z}))$

Figure 4: Comparison of MRI reconstruction: (a) The normally trained network $f_{\boldsymbol{\theta}_{\mathrm{NT}}}$ at the benign input, (b) the AT model $f_{\boldsymbol{\theta}_{\mathrm{AT}}}$ at the adversarially perturbed input, (c) the AT model $f_{\boldsymbol{\theta}_{\mathrm{AT}}}$ at the transformed input (by rotation of $180°$). we rotate it $180°$ for easier comparison with others.

**Generalized AT (GAT) with data augmentation**    Spurred by the lack of robustness of AT against input transformations, we ask if the transformed data can be infused into the AT framework. Meanwhile, such integration of AT with data augmentation is expected to be as simple as possible without needing additional data annotations. To this end, we can augment the training data set $\mathcal{D}$ with the newly paired transformed data $\tilde{\mathcal{D}} = \{T(\mathbf{z}), T(\mathbf{x}^*)\}_{(\mathbf{z},\mathbf{x}^*)\in\mathcal{D}}$. In this context, we will generalize AT into the following regularized optimization framework

$$\min_{\boldsymbol{\theta}} \ \underbrace{\mathbb{E}_{(\mathbf{z},\mathbf{x}_*)\in\mathcal{D}\cup\tilde{\mathcal{D}}}\left[\ell(f_{\boldsymbol{\theta}}(\mathbf{z}),\mathbf{x}^*)\right]}_{\text{augmented normal training}} + \gamma \ \underbrace{\mathbb{E}_{(\mathbf{z},\mathbf{x}_*)\in\mathcal{D}\cup\tilde{\mathcal{D}}}\max_{\|\boldsymbol{\delta}\|_\infty\leq\epsilon}\left[\ell(f_{\boldsymbol{\theta}}(\mathbf{z}+\boldsymbol{\delta}),f_{\boldsymbol{\theta}}(\mathbf{z}))\right]}_{\text{robustness regularization}}, \tag{8}$$

where $\gamma > 0$ is a regularization parameter to strike a balance between normal training and robust training. Different from (7), we explicitly introduce an augmented standard training objective to improve the reconstruction accuracy when facing input transformations, and we introduce a 'label-free' robust regularization that penalizes the discrepancy between the normal reconstruction and the adversarial reconstruction. This robustness regularization is commonly used to tradeoff the standard model accuracy and the model robustness [Zhang et al., 2019].

To solve (8), one major difficulty is how to select the proper data transformation operations $T(\cdot)$ to construct $\tilde{\mathcal{D}}$. Motivated by the success of data augmentation in improving distributional robustness of image classification [Geirhos et al., 2018, Hendrycks et al., 2019], we consider the following three representative data transformation operations: rotation [Taylor and Nitschke, 2018], cutout [DeVries and Taylor, 2017], and cutmix [Yun et al., 2019]; see Fig. A1 for an illustrative example. The rationale behind these augmentation operations is that invariant deep features used for image reconstruction can be learned against spatial perturbations, *e.g.* , occlusion, translation, and rotation. Our experiments in Sec. 5 will show that the combination of these augmentation operations can lead to a superior robustness to input transformations. Meanwhile, the model learned from (8) (denoted by $\boldsymbol{\theta}_{\mathrm{GAT}}$) also maintains robustness against adversarial input perturbations like AT (see Fig. 5).

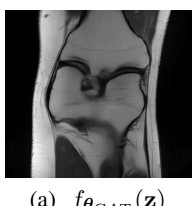
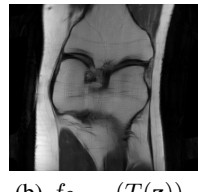
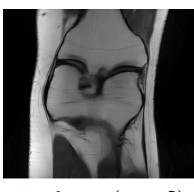

(a) $f_{\boldsymbol{\theta}_{\mathrm{GAT}}}(\mathbf{z})$      (b) $f_{\boldsymbol{\theta}_{\mathrm{GAT}}}(T(\mathbf{z}))$      (c) $f_{\boldsymbol{\theta}_{\mathrm{GAT}}}(\mathbf{z} + \boldsymbol{\delta})$

Figure 5: Example of MRI reconstruction using GAT, which leads to the satisfactory reconstruction performance against the benign input $\mathbf{z}$ as well as the perturbed inputs $T(\mathbf{z})$ and $\mathbf{z} + \boldsymbol{\delta}$.we rotate output at $T(\mathbf{z})$ $180°$ for easier comparison with others.

## 5 Experiments

### 5.1 Experiment Setup

**Datasets and model architecture**    In this paper, we focus on the Fully-sampled coronal proton density knee MRI dataset with 15-channel coils, which were obtained from the fastMRI database [Knoll et al., 2020b]. The image size is $320 \times 368$. In our paper, we used the unrolling-based image reconstruction model MoDL [Aggarwal et al., 2018].

**Training setup**    All of our models were trained in an end-to-end fashion using the Adam optimizer with learning rate $1 \times 10^{-4}$ and batch size 1. When implementing AT and GAT, we used the 10-step

PGD attack [Madry et al., 2017] with $\epsilon = 0.03/255$ for solving the inner maximization problem. When generating the transformed dataset $\tilde{\mathcal{D}}$ in GAT, we sampled one transformation operation for each original data sample from the random cutout, random cutmix, and rotate 180° uniformly. And we choose the regularization parameter $\gamma = 3$ in the implementation of GAT (8).

**Evaluation metrics** In our paper, we used both normalized mean squared error (NMSE) and structural similarity index measure (SSIM) to evaluate the quality of reconstructed images. The higher SSIM signifies higher quality reconstruction results, while the higher NMSE represents the lower quality of image reconstruction. To evaluate the robustness and accuracy of a model, we evaluate its performance when facing three types of test-time inputs: **1) Benign test input z**, which corresponds to the standard reconstruction accuracy; **2) Adversarial test input** $(\mathbf{z} + \boldsymbol{\delta})$, which is generated using the 20-step PGD attack with $\epsilon = 0.03/255$ and corresponds to the adversarial robustness; **3) Transformed test input** $T(\mathbf{z})$: model's robustness against input spatial transformations.

## 5.2 Experiment results

**Reconstruction accuracy and robustness** We compared our proposed GAT with NT and AT. The overall quantitative results are shown in Tab. 1. As we can see, GAT is the most robust model against spatial transformation-based perturbations, without sacrificing much on accuracies against adversarial test inputs and benign test inputs. These suggest that our proposed GAT improves robustness against adversarial perturbation and spatial transformation-based perturbation with a slight standard accuracy drop.

Table 1: NMSE and SSIM results achieved by differently trained models, including NT (normal training), AT (adversarial training), and GAT (ours). The relative improvement/degradation with respect to NT is shown.

| MRI reconstruction results on FastMRI | | | | | | |
|---|---|---|---|---|---|---|
| Method | benign test input (z) | | adversarial test input (z + δ) | | transformed test input (T(z)) | |
| | NMSE | SSIM | NMSE | SSIM | NMSE | SSIM |
| NT | **0.0013** | **0.9442** | 0.4952 | 0.4392 | 0.6824 | 0.7577 |
| AT | +0.0014 | -0.0478 | **-0.4900** | **+0.3827** | -0.5530 | +0.0354 |
| GAT | +0.0017 | -0.0521 | -0.4892 | +0.3748 | **-0.6619** | **+0.0790** |

**Effect of data augmentations** Next, we investigate how the choice of data augmentation affects model robustness. Tab. 2 shows the NMSE and SSIM performance of our approach (GAT) using different data augmentation strategies: cutout-only, cutmax-only, rotation-only, and their combination. As we can see, each transformation operator used in GAT can improve the robustness against the spatial input transformations. However, the integration of all three transformation types into GAT achieves the best robustness at transformed test inputs. Meanwhile, we find that the use of cutout-only is able to achieve a great tradeoff with the improvement in adversarial robustness.

Table 2: NMSE and SSIM results achieved by GAT under different data augmentation methods. The relative performance is shown with respect to NT.

| Comparison of different augmentation methods | | | | | | |
|---|---|---|---|---|---|---|
| Method | benign test input (z) | | adversarial test input (z + δ) | | transformed test input (T(z)) | |
| | NMSE | SSIM | NMSE | SSIM | NMSE | SSIM |
| NT | **0.0013** | **0.9442** | 0.4952 | 0.4392 | 0.6824 | 0.7577 |
| GAT(cutout) | +0.0014 | -0.0502 | **-0.4896** | **+0.3824** | -0.6262 | +0.0777 |
| GAT(cutmix) | +0.0014 | -0.0526 | -0.4894 | +0.3731 | -0.6183 | +0.0668 |
| GAT(rotation) | +0.0028 | -0.0535 | -0.4877 | +0.3180 | -0.6593 | +0.0476 |
| GAT | +0.0017 | -0.0521 | -0.4892 | +0.3748 | **-0.6619** | **+0.0790** |

**Visualization of reconstructed MRI images** Fig. A2 shows examples of reconstructed MRI images using different approaches when facing different types of a test-time examples. For ease of comparison, we apply the inverse spatial transformation to the reconstruction result of a spatially-transformed input. As shown in Fig. A2-(b), (d), and (e), there exist less visible artifacts in the reconstruction results of our GAT-trained model. Compared to NT, both AT and GAT significantly improves model robustness against adversarial perturbations. While Fig. A2-(a), (d), and (g) suggest that there might exist an accuracy-robustness tradeoff.

## 6 Conclusion

In this paper, we investigate the problem of adversarial robustness of DL-based MRI image reconstruction. We show that a deep image reconstruction model is highly susceptible to not only pixel-level adversarial perturbations but also input spatial transformations, e.g., cropping and rotation. To improve the resilience of MRI image reconstruction, we propose a generalized adversarial training method that is built upon a new robustness regularization metric by taking into account both adversarial perturbations and data transformations. We show that our proposal is effective in a fastMRI dataset by comparing with state-of-the-art baselines under a variety of evaluation metrics. In the future, we would like to generalize our approach to the self-supervised learning paradigm, and improve its computation efficiency through more lightweight robust training protocols.

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

# A Appendix

## A.1 Transformation operations

In this section, we provide some illustrations for different data transformation operations.

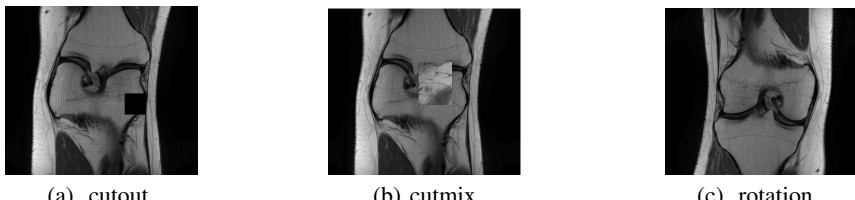

(a) cutout  (b) cutmix  (c) rotation

Figure A1: Examples for different data transformation operations.

## A.2 Results visualization

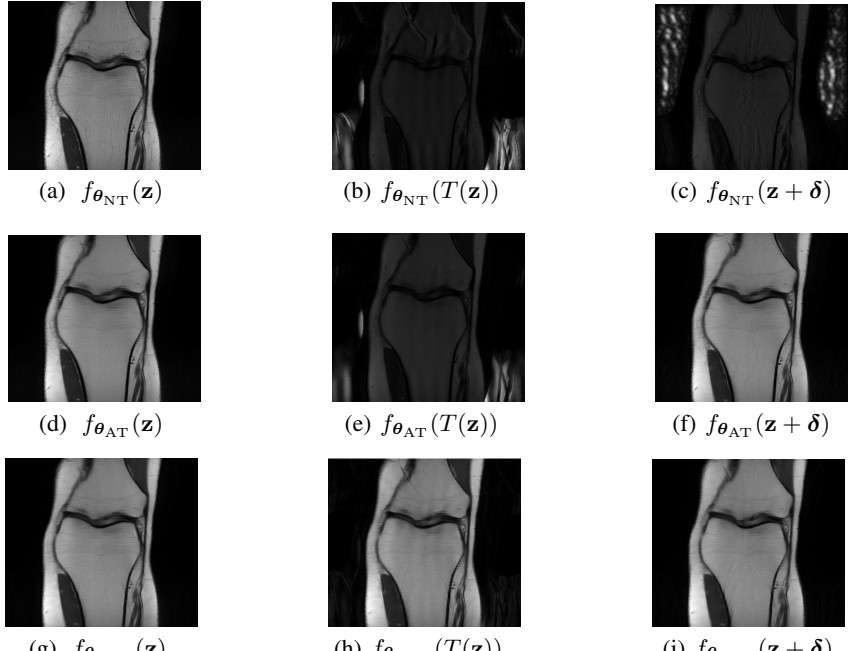

(a) $f_{\boldsymbol{\theta}_{\mathrm{NT}}}(\mathbf{z})$  (b) $f_{\boldsymbol{\theta}_{\mathrm{NT}}}(T(\mathbf{z}))$  (c) $f_{\boldsymbol{\theta}_{\mathrm{NT}}}(\mathbf{z}+\boldsymbol{\delta})$

(d) $f_{\boldsymbol{\theta}_{\mathrm{AT}}}(\mathbf{z})$  (e) $f_{\boldsymbol{\theta}_{\mathrm{AT}}}(T(\mathbf{z}))$  (f) $f_{\boldsymbol{\theta}_{\mathrm{AT}}}(\mathbf{z}+\boldsymbol{\delta})$

(g) $f_{\boldsymbol{\theta}_{\mathrm{GAT}}}(\mathbf{z})$  (h) $f_{\boldsymbol{\theta}_{\mathrm{GAT}}}(T(\mathbf{z}))$  (i) $f_{\boldsymbol{\theta}_{\mathrm{GAT}}}(\mathbf{z}+\boldsymbol{\delta})$

Figure A2: Visualization of MRI reconstructed images. Row 1: NT; Row 2: AT; Row 3: GAT.

