# OpenReview forum: "On the Robustness of deep learning-based MRI Reconstruction to image transformations"
_NeurIPS.cc/2022/Workshop/TSRML — TSRML2022_

### Official Review · Reviewer_LpAh · 2022-10-13
**Well-written paper applying adversarial training techniques to a real-world robustness application**

**Overall Rating:** 6

**Summary:**

This paper studies adversarial robustness and robustness to spatial transformations for deep image reconstruction, specifically magnetic resonance imaging (MRI). The authors show the vulnerability of deep MRI image reconstruction models to norm-bounded adversarial perturbations and spatial transformations by observing a decrease in the structural similarity index measure (SSIM) and an increase in the normalized mean squared error (NMSE) between the reconstructed image and the ground truth image when the sub-sampled input image is perturbed. This paper proposes using TRADES [Zhang et al., 2019] adversarial training with data augmentation to improve the quality of the reconstructed images in the face of adversarial or spatial perturbations. The authors evaluate this approach compared to standard training on an MRI dataset.

**Strengths:**

This paper is well-written and easy to follow. The authors apply adversarial techniques to the less commonly studied domain of image reconstruction. The specific application to MRI is an interesting real-world example. The proposed use of adversarial training with data augmentation results in better image reconstructions than standard training on adversarially and spatially perturbed input images.

**Weaknesses:**

From what I understand, the proposed training method is TRADES adversarial training applied to image reconstruction instead of image classification, plus data augmentation, so I’d recommend just calling your method TRADES rather than GAT. Given that the main contribution of your paper is the application of adversarial training to a real world problem, I’d consider emphasizing/adding more content about the potential dangers of degraded MRI reconstructions. More on this point, it’d be beneficial to add some context with regards to the practicality of encountering adversarial/spatial transformations in these settings.

Additional comments:
- I would think about adding the NMSE or SSIM scores to the relevant images in Figures 2-5 to quantitatively illustrate the differences.
- The choice of epsilon=0.03/255 seems quite small?
- The tables are a little too small in my opinion.


**Overall Recommendation:**

Overall this paper is well written, and studies adversarial robustness on a real-world dataset/application. The paper uses existing adversarial methods to show better results as compared to standard training, which is expected, and could be improved by adding more context with regards to the specific MRI problem.

**Review Confidence:**

4: The reviewer is confident but not absolutely certain that the evaluation is correct

---

### Official Review · Reviewer_EfWN · 2022-10-20
**A method to enhance the adversarial robustness of models in MRI construction**

**Overall Rating:** 6

**Summary:**

This paper focuses on the adversarial robustness of model in the task of image reconstruction. The authors also find the vulnerability of models to transformations on the input. Then, they propose a method to boost the robustness of the model against both adversarial perturbations and image transformations. Experimental results have shown the effectiveness of their method.

**Strengths:**

- This paper focuses on the robustness of models in the image reconstruction task, and proposes a new method to improve the adversarial robustness.
- The authors do not only consider the adversarial robustness, but also notice the vulnerability of the model against image transformations.
- Experiments have shown the effectiveness of the proposed method.

**Weaknesses:**

- In Eq. (5), why is the loss computed by using $f_{\theta_{NT}}(z)$ instead of the ground truth $x^*?$
- It would be better to compare the performance on adversarial examples of augmented images (i.e., $T(x)+\delta$) in experiments, because such images are also used in training of Eq. (8).
- It would be better is the authors could investigate whether some augmentations (e.g., cutout) are more useful than other transformations (e.g., cutmix) in GAT.

**Overall Recommendation:**

Overall this is a good submission. The authors propose a method to improve the robustness of models in image reconstruction, and experiments demonstrate the effectiveness of the proposed method.

**Review Confidence:**

5: The reviewer is absolutely certain that the evaluation is correct and very familiar with the relevant literature

---

### Official Review · Reviewer_KwYf · 2022-10-20
**Well-written in general but unclear motivation and novelty**

**Overall Rating:** 5

**Summary:**

The authors investigate the robustness of deep-learning (DL) based MRI image reconstruction with respect to spatial transformations to the input image. They claim that the proposed general adversarial training (GAT) with training dataset augmented with spatially transformed images provides the best reconstruction performance especially when the input image is subject to adversarial perturbation and spatial transformation.

**Strengths:**

The paper is well-written and easy to understand. The background on MRI image reconstruction is sufficient. They discover that DL based MRI image reconstruction method is sensitive to spatial transformation of the input image.

**Weaknesses:**

It seems that the novelty of the paper lies in the discovery of sensitivity to spatial transformation and consequent resolution using a combination of adversarial training and dataset augmentation which are well-known in literature. Furthermore, it is not clear whether spatial transformation is an issue unique to MRI images or somehow a physical artifact while obtaining images from the MRI machine.
If the issue of spatial transformation is not unique to MRI images, it is difficult to ascertain the novelty of this paper solely on the basis of discovery of sensitivity, given that dataset augmentation is not new.

Some more comments:
1. Why are some NMSE values negative in Tables 1 and 2? It is a non-negative value unless normalized by a negative number. Please define the metric used to obtain these results.
2. What is SSIM? Please define it in the paper.
3. I do not see how Fig A2-(b), (d), (e) convey that GAT-trained model based reconstruction has less visible artifacts.
4. Why do Fig. A2(a), (d), (g) suggest accuracy-robustness tradeoff? Please elaborate.


**Overall Recommendation:**

Although the idea is interesting, the novelty of the paper is not clear as it seems to be addressing standard problems in image reconstruction using standard methods of adversarial training and dataset augmentation in DL-based methods without sufficient motivation as to what makes the proposed method unique to MRI images.

**Review Confidence:**

4: The reviewer is confident but not absolutely certain that the evaluation is correct

---

### Decision · Program_Chairs · 2022-10-23

**Decision:**

Accept

**Comment:**

While the technical contribution of this paper is limited, it studies a practical problem in detail with good empirical results, and the results will be useful for researchers in medical domains.